# AIPI: Network Status Identification on Multi-Protocol Wireless Sensor Networks

**DOI:** 10.3390/s25051347

**Published:** 2025-02-22

**Authors:** Peng Jiang, Xinglin Feng, Renhai Feng, Junpeng Cui

**Affiliations:** 1School of Electrical and Information Engineering, Weijin Road Campus, Tianjin University, Nankai District, Tianjin 300072, China; jiangpeng@motimo.com.cn (P.J.); fengxinglin@tju.edu.cn (X.F.); 2Tianjin Motimo Membrane Tech. Co., Ltd., Tianjin 300457, China; 3Software and Communication School, TianjinSino-German University of Applied Sciences, Tianjin 300350, China; cuijunpeng@tsguas.edu.cn

**Keywords:** topology identification, active interfere, frequency hopping, passive interception, Granger causality

## Abstract

Topology control is important for extending networks lifetime and reducing interference. The accuracy of topology identification plays a crucial role in topology control. Traditional passive interception can only identify the connectivity among cooperative sensor networks with known protocol. This paper proposes a novel method called Active Interfere and Passive Interception (AIPI) to identify the topology of non-cooperative sensor networks by using both active and passive interceptions. Active interception uses full duplex sensors to disrupt communication until frequency hopped to acquire distance information, and thus, infer their connectivity and calculate the location after modifying error in a non-cooperative sensor network. Passive interception uses Granger causality to infer the connectivity between two communication nodes after getting the time frame structure in physical layer. Passive interception is applied to conserve power consumption after obtaining physical information via active interception. Simulation results indicate that AIPI can identify the topology of non-cooperative sensor networks with a higher accuracy than traditional method.

## 1. Introduction

Wireless Sensor Networks (WSN) consisting of a large number of self-organizing nodes are widely deployed in environmental monitoring, smart homes, and healthcare. In these applications, network topology plays a crucial role. Topology not only affects capacity and energy consumption but also directly impacts the data collection and reliability of the network. Some algorithms including routing algorithms [1], data aggregation algorithms, load balancing, and some data aggregation algorithms [2] need to know the topology to ensure optimal performance. Therefore, identifying the topology of WSN is meaningful for maintaining their operation. Traditional passive interception methods include the correlation-based method [3], information-theory-based method [4], Granger-causality-test-based method [5], compression-sensing-based method [6], drive-response-based method [7], and graph-based neural network method [8]. Passive interception obtains the connections between nodes by capturing and decoding data packets, and it is susceptible to channel noise. The existing literature demonstrates that passive interception can identify the topology of sensor network assuming known protocol [3,4,5,6,7,8], in which the type and structure of the data package is known. Unlike cooperative WSN, the communication protocols of non-cooperative WSN are unknown. Such assumption limited their application scope under non-cooperative WSN.

In response to non-cooperative networks, Xu et al. [9] first introduced the concept of active interception, utilizing full-duplex on interception device. This method can interfere signal reception of the suspicious receiver, thus intercepting suspicious information, and thereby, enhancing eavesdropping. Subsequently, some scholars have further applied the strategy of Xu. Hui et al. [10] studied the problem of maximizing the average monitoring rate of a lawful monitoring system which actively intercepts on suspicious wireless links through interfering Rayleigh fading channels. Yunlong et al. [11] investigated millimeter-wave information monitoring systems, where suspicious transmitters in the network send messages to suspicious receivers under the supervision of a monitoring controller, achieving lawful surveillance of suspicious links. In order to increase the probability of successful intercepting, Yitao et al. [12] proposed cognitive interference to alter the long-term trust of the suspicious links under parallel channels, thereby encourage the links to transmit on a smaller subset of unblocked channels with a lower transmission rate. Hu et al. [13] investigated active intercepting of half-duplex lawful monitor in a wireless power supply multi-channel suspicious system, where the suspicious transmitters harvest energy from power beacons and then communicate with the suspicious destination through parallel channels. Additionally, single-hop relay networks [14], unmanned aerial vehicle (UAV)-assisted downlink multicast [15], broadcast and uplink multi-access [16], multi-user communication [17], collaborative monitoring systems [18], and intelligent reflecting surface-assisted intercepting networks [19] all applied active interception. Compare to passive interception, active interception exchange the information through interfering actively, which avoids decoding data package. In order to identify the topology of sensor network accurately, combining localization and active interception is a promising direction. The location of the communication nodes provide extra information about topology. Existing localization algorithm include range-dependent [20,21,22] and range-independent [23,24]. These techniques often require additional hardware and consume significant resources. Recent advancements in sensing technologies [25] and distributed systems [26,27] offer potential solutions to these challenges. To address the limitations of these methods, the level of the power of signal from interfering device can used to infer position information (PI).

Therefore, this paper combines active interception and local information extraction to identify the topology of sensor network with unknown communication protocol and use passive interception with the Granger causality test to identify the topology of sensor network while the communication protocol is decoded. The topology sensing model shown in Figure 1 is considered in this paper, where three full duplex sensors transmit the information from target WSN to fusion center. Fusion center can apply active interception or passive interception in different stage. Figure 2 illustrates the overall framework of the AIPI method. In the initial stage, the AIPI method applies active interception to identify topology and obtain physical layer information. To conserve power consumption, the AIPI method applies passive interception to track dynamic topology constantly.

The three main contribution of this paper are as follows:(1)Active interception identifies topology through interfering node communication with an energy-effective dichotomy;(2)Local information extraction with self-adaptive error distribution provides extra effective information for active interception;(3)Passive interception with the Granger causality test is proposed to track topology of WSN with physical layer information.

Traditional passive interception such as the Granger causality test [28] and Hawkess process [29] have advantages regarding power consumption because they only need to sense the signal. However, they are susceptible to noise and are only applicable to cooperative WSN. AINL [30] is applicable to non-cooperative and cooperative WSN, but it can only extract the node PI with dichotomy in a small noise environment. AIPI can not only extract node PI with energy effective dichotomy, but it is also insensitive to noise in non-cooperative WSN, which means that it has superior performance on universality, ennergy efficience, and anti-interference. AIPI applies active and passive interception at different stages, causing higher power consumption than pure passive interception and less than pure AINL. A comparison of AIPI with existing works on topology inference is shown in Figure 3.

The rest of this paper is organized as follows. In Section 2, we introduce the use of active interference for local information extraction, and explain self-adaptive error correction mechanism in our proposed method using the maximum likelihood function. Finally, we apply passive interception to infer topology based on Granger causality. The simulation results are given in Section 3 and the conclusions for the work are provided in Section 4.

## 2. Theoretical Basis

### 2.1. Problem Formulation

In mathematical notation, a network graph can be represented by g={v,σ}, where v={1,2,⋯N} denotes the set of vertices, *N* is the number of elements in *v*, that is, the number of nodes in the network, and σ denotes the set of edge. The element in the adjacency matrix A of the network can be represented as (Equation 1),(1)$$a_{i,j}=\left\{\begin{array}{c} 1,\;(i,j)\in \sigma \\ 0,\;(i,j)\notin \sigma \end{array}\right.$$
where i,j∈N. We formulate the topology inference problem as (Equation 2),(2)$$\min∥X^{-}{-X∥}_{2}+{∥A^{-}-A∥}_{1}$$
where $X^{-}$ represents estimated node coordinate, *X* represents real node coordinate, and $A^{-}$ represents the inference topology.

### 2.2. Active Interception

We use active interception to identify the topology of WSN with unknown protocol. Assuming WSN has N communication nodes, while all the nodes can transmit and 20% nodes can receive. In the target network, different node pairs communicate using different frequency bands, preventing collisions. Three full duplex sensors locate the target node by colliding on the corresponding frequency bands. As shown in Figure 4, T denotes the transmitter, R denotes the receiver, and EVE nodes represent three interfering nodes. T and R are communicating at X band, and then three full duplex sensors EVE1, EVE2, and EVE3 send interference signals with different power levels until the frequency hopped in communication. PI and connectivity are deduced through trilateration after transforming power level into distance. We can distinguish different communication node when the error between the real coordinate and the estimated coordinate is less than a threshold.

Algorithm 1 explains the strategy of power adapting for EVE. Define the upper and lower bounds on EVE transmission pseudo-noise power as $P_{\max}$ and $P_{\min}$, the transmission distance is bounded by $R_{\max}$ and $R_{\min}$. Define the $R_{\mathrm{real}}$ to be the real distance between R and EVE. Change power of EVE $R_{1}$ through dichotomy coefficient until the error between $R_{1}^{\max}$ and $R_{1}^{\min}$ is less than error δ. Apply same strategy to T can also obtain the coordinate.

The error of distance between R and EVE ϵ will inevitably occur when using the dichotomy to calculate it. The ϵ lead to the three circles cannot intersect at a single point based on trilateration as shown in Figure 5. The error is actually subtle, and thus, for interpreting, we enlarge it. In this case, we consider the midpoint (green) of the line connecting the two closest intersection points (red) as the result of local information extraction.
**Algorithm 1** Interference Strategy of Active Interception.  1:**Input:** dichotomy coefficient α, error δ, $R_{\mathrm{real}}$  2:**Initialization:** $R_{1}^{\min}=R_{\min}$, $R_{1}^{\max}=R_{\max}$  3:Deploy transmitting node, receiving node, and EVE  4:EVE transmits interference signal  5:**while** $R_{1}^{\max}-R_{1}^{\min}>\delta$ **do**  6:    $R_{1}=R_{1}^{\min}+\alpha \left(R_{1}^{\max}-R_{1}^{\min}\right)$  7:    **if** $R_{1}<R_{\mathrm{real}}$ **then**  8:        set $R_{1}^{\min}=R_{1}$  9:    **else**10:        set $R_{1}^{\max}=R_{1}$11:    **end if**12:**end while**13:Calculate node coordinate with Formula (Equation 4)14:**Output:** (x,y)

It is impossible to extract the local information with a very small error δ because of the environment noise. To extract the local information with a bigger error δ accurately, we deploy data from train sample with same strategy to extract local information of unknown WSN with self-adaptive error. Assuming a WSN whose PI is known, we use (x,y) represent the result of local information extraction through trilateration and $\left(x^{\prime },y^{\prime }\right)$ represent the real PI, so that the error between them can be denoted as $\left(\Delta x,\Delta y\right)=\left(x^{\prime }-x,y^{\prime }-y\right)$. Calculate the errors of coordinates for all communication nodes. Since the coordinates of the nodes are represented in two dimensions, we use two Gaussian functions to fit the obtained errors through Algorithm 2.

The Gaussian Mixture Models can be represented as a linear combination of Gaussian distributions as shown in (Equation 3):(3)$$p\left(x\right)=\sum_{k=1}^{K}\pi_{k}N\left(x|u_{k},\Sigma_{k}\right),$$
where N follows a Gaussian distribution, $u_{k}$ and $\Sigma_{k}$ are the mean and covariance matrix corresponding of the Gaussian distribution, and $\pi_{k}$ is the probability of each Gaussian distribution with $\pi_{k}$ satisfying $\sum_{k=1}^{K}\pi_{k}=1$. Solve the variable of Gaussian distribution $n_{1}∼N\left(u_{1},\Sigma_{1}\right)$ fitting Δx and $n_{2}∼N\left(u_{2},\Sigma_{2}\right)$ fitting Δy through Expectation-Maximization (EM) algorithm.
**Algorithm 2** EM Algorithm for Gaussian Mixture Model Parameter Estimation  1:**Input:** Observed dataset *X*, number of components *k*, convergence threshold η  2:**Initialization**: Randomly initialize each Gaussian model parameter $\theta_{k}$ (the mean $\mu_{k}$, covariance matrix $\Sigma_{k}$, and weights $\pi_{k}$ for each Gaussian component), ensuring that $\sum_{k}\pi_{k}=1$  3:**while** $\theta^{t}-\theta^{t-1}>\eta$ **do**  4:    **E-Step (Expectation)**:  5:    **for** each observation $x_{n}$ **do**  6:        Compute the posterior probability that $x_{n}$ belongs to each Gaussian component:$$p\left(z_{nk}|x_{n}\right)=\frac{\pi_{k}\cdot N\left(x_{n}|\mu_{k},\Sigma_{k}\right)}{\sum_{j}\pi_{j}\cdot N\left(x_{n}|\mu_{j},\Sigma_{j}\right)}$$  7:    **end for**  8:    **M-Step (Maximization)**:  9:    **for** each Gaussian component *k* **do**10:        Update the weight:$$\;\pi_{k}=\frac{N_{k}}{N},\;N_{k}\;\mathrm{is}\;\mathrm{the}\;\mathrm{number}\;\mathrm{of}\;\mathrm{data}\;\mathrm{points}\;\mathrm{belonging}\;\mathrm{to}\;\mathrm{the}\;k-\mathrm{th}\;\mathrm{component}$$11:        Update the mean:$$\mu_{k}=\frac{\sum_{n}p\left(z_{nk}|x_{n}\right)\cdot x_{n}}{\sum_{n}p\left(z_{nk}|x_{n}\right)}$$12:        Update the covariance matrix:$$\Sigma_{k}=\frac{\sum_{n}p\left(z_{nk}|x_{n}\right)\cdot \left(x_{n}-\mu_{k}\right){\left(x_{n}-\mu_{k}\right)}^{T}}{\sum_{n}p\left(z_{nk}|x_{n}\right)}$$13:    **end for**14:**end while**15:**Output:** the mean $\mu_{k}$, covariance matrix $\Sigma_{k}$, and weights $\pi_{k}$ for each Gaussian component

For acquiring PI of an unknown WSN, apply the parameter Σ and *u* to extract position information. The coordinates of R is $\left(x_{n},y_{n}\right)$. The coordinates of EVE are $\left(x_{i},y_{i}\right)$, and the distance between R and EVE are $r_{i}$, where i=1,2,3. Construct a set of trilateration equations as shown in (Equation 4):(4)$$\left\{\begin{array}{c} r_{1}^{2}={\left(x_{1}-x_{n}\right)}^{2}+{\left(y_{1}-y_{n}\right)}^{2}\\ r_{2}^{2}={\left(x_{2}-x_{n}\right)}^{2}+{\left(y_{2}-y_{n}\right)}^{2}\\ r_{3}^{2}={\left(x_{3}-x_{n}\right)}^{2}+{\left(y_{3}-y_{n}\right)}^{2} \end{array}\right.$$
Derive (Equation 5) from (Equation 4):(5)$$\left\{\begin{array}{c} r_{2}^{2}-r_{1}^{2}=x_{2}^{2}-x_{1}^{2}-2\left(x_{2}-x_{1}\right)x_{n}+y_{2}^{2}-y_{1}^{2}-2\left(y_{2}-y_{1}\right)y_{n}\\ r_{3}^{2}-r_{1}^{2}=x_{3}^{2}-x_{1}^{2}-2\left(x_{3}-x_{1}\right)x_{n}+y_{3}^{2}-y_{1}^{2}-2\left(y_{3}-y_{1}\right)y_{n} \end{array}\right.$$
Transform (Equation 5) into matrix to obtain (Equation 6),(6)$$\left[\begin{array}{cc} x_{2}-x_{1} & y_{2}-y_{1}\\ x_{3}-x_{1} & y_{3}-y_{1} \end{array}\right]\cdot \left[\begin{array}{c} x_{n}\\ y_{n} \end{array}\right]=\left[\begin{array}{c} -\frac{1}{2}k_{2}^{2}+\frac{1}{2}r_{2}^{2}-\frac{1}{2}r_{1}^{2}\\ -\frac{1}{2}k_{3}^{2}+\frac{1}{2}r_{3}^{2}-\frac{1}{2}r_{1}^{2} \end{array}\right]$$
where $k_{i}=x_{i}^{2}-x_{1}^{2}+y_{i}^{2}-y_{1}^{2}$, and the matrix equation can be simplified to represent as HX=Y. Using (Equation 6) and the two Gaussian functions obtained from fitting the errors, construct (Equation 7):(7)$$Y=HX+n_{1}+n_{2}$$
Derive (Equation 8) from (Equation 7):(8)$$Y-HX-n_{2}=n_{1}$$
Since $n_{1}$ follows a Gaussian distribution, $Y-HX-n_{2}$ also follows a Gaussian distribution with the same parameters. Let *M* represent $Y-HX-n_{2}$, then the distribution of *M* is expressed as (Equation 9),(9)$$N\left(M|u_{1},\Sigma_{1}\right)=\frac{1}{2\pi }\frac{1}{|\Sigma_{1}{|}^{1/2}}\exp\left\{-\frac{1}{2}{\left(M-u_{1}\right)}^{T}\Sigma_{1}^{-1}\left(M-u_{1}\right)\right\}$$
of which log-likelihood function, after taking the logarithm, can be expressed as (Equation 10):(10)$$\begin{array}{cc} \; & \ln p(M|u_{1},\Sigma_{1})=\\ \; & -\ln\left(2\pi \right)-\frac{1}{2}\ln\left|\Sigma_{1}\right|-\frac{1}{2}{\left(M-u_{1}\right)}^{T}\Sigma_{1}^{-1}\left(M-u_{1}\right) \end{array}$$
Opening the ${\left(M-u_{1}\right)}^{T}\Sigma_{1}^{-1}\left(M-u_{1}\right)$, we can obtain (Equation 11):(11)$$\begin{array}{cc} \; & {\left(Y-HX-n_{2}-u_{1}\right)}^{T}\Sigma_{1}^{-1}\left(Y-HX-n_{2}-u_{1}\right)\\ \; & =Y^{T}\Sigma_{1}^{-1}Y-Y^{T}\Sigma_{1}^{-1}HX-Y^{T}\Sigma_{1}^{-1}n_{2}-Y^{T}\Sigma_{1}^{-1}u_{1}\\ \; & -X^{T}H^{T}\Sigma_{1}^{-1}Y+X^{T}H^{T}\Sigma_{1}^{-1}HX\\ \; & +X^{T}H^{T}\Sigma_{1}^{-1}n_{2}+X^{T}H^{T}\Sigma_{1}^{-1}u_{1}\\ \; & -n_{2}^{T}\Sigma_{1}^{-1}Y+n_{2}^{T}\Sigma_{1}^{-1}HX+n_{2}^{T}\Sigma_{1}^{-1}n_{2}+n_{2}^{T}\Sigma_{1}^{-1}u_{1}\\ \; & -u_{1}^{T}\Sigma_{1}^{-1}Y+u_{1}^{T}\Sigma_{1}^{-1}HX+u_{1}^{T}\Sigma_{1}^{-1}n_{2}+u_{1}^{T}\Sigma_{1}^{-1}u_{1} \end{array}$$
Since $n_{2}$ is independent, derive (Equation 12) from (Equation 10) and (Equation 11):(12)$$\begin{array}{cc} \; & \frac{\partial \ln p\left(M|u_{1},\Sigma_{1}\right)}{\partial X}=\\ \; & H^{T}\Sigma_{1}^{-1}HX+{\left(H^{T}\Sigma_{1}^{-1}H\right)}^{T}X+H^{T}\Sigma_{1}^{-1}u_{1}\\ \; & +H^{T}{\Sigma_{1}^{-1}}^{T}u_{1}-H^{T}{\Sigma_{1}^{-1}}^{T}Y-H^{T}\Sigma_{1}^{-1}Y \end{array}$$
Let Equation (Equation 12) equal zero, solve for *X*, and thus, obtain the PI. Distinguishing the different communication nodes in WSN, we can identify the topology of WSN.

### 2.3. Passive Interception

After active interception distinguish communication mode in physical layer, we apply steadily passive interception to monitor the topology of WSN in a long time, which can conserve energy effectively. Assume the WSN applies Transmission Control Protocol/Internet Protocol (TCP/IP) and Automatic Repeat-reQuest (ARQ) protocol, which means that communication node will receive an Acknowledge character package (ACK) from the other communication node after transmitting data package successfully. The EVE captures the package from WSN at a period T and then acquire its transmission time and transmitter information. Divide period T time into L slots to acquire the time series. In the IEEE 802.15.4 standard, the baud rate is 250 kbps. The time slot length *t* should be less than the data transmission time $T_{i}=L_{data}/Br$, where $L_{data}$ is the data package length and the Br is the baud rate. The optimal number of slots *L* can be obtain through $L=T/L_{s}$. Considering the WSN is sparse and undirected, apply the Granger causality to infer two time series from two communication nodes whether exist causality relationship and then identify the connectivity. We can identify the topology of WSN after deducing all pairs of communication if there exists connectivity.

For each node i∈v, a time series is generated when the node transmits a data package and an ACK package. More precisely, for the set of packet transmitting times $T_{i}$, the binary time series $x_{i,I}$ is represented as (Equation 13),(13)$$x_{i,I}=\left\{\begin{array}{c} 1,IT_{s}<T_{i}<\left(L-I\right)T_{s}\\ 0,\mathrm{others} \end{array}\right.$$
where *i* represents the node ID, *I* indicates the time slot index, *L* is the total number of time slots, and $T_{s}$ represents the length of the time slots. The schematic diagram of the node’s time series can be represented as shown in Figure 6.

Granger causality is used to compare prediction error to determine whether two variables are related. For the time series $X=\{x_{1},x_{2},x_{3},\ldots \}$ and $Y=\{y_{1},y_{2},y_{3},\ldots \}$, if the knowledge of past values of Y contributes some additional information about $x_{n}$ more than the knowledge of just past of *X*, we consider that *Y* Granger causes *X*. Granger causality is always modeled as a hypothesis test problem, where the null hypothesis $H_{0}$ means that *Y* does not Granger cause *X* and the alternative hypothesis $H_{1}$ means that Y Granger *X*. The hypothesis test is represented as (Equation 14) and (Equation 15),(14)$$H_{0}:X\left[n\right]=\sum_{i=1}^{k}a_{i}X\left[n-i\right]+\varepsilon \left[n\right]$$(15)$$H_{1}:X\left[n\right]=\sum_{i=1}^{k}b_{i}X\left[n-i\right]+\sum_{i=1}^{k}c_{i}Y\left[n-i\right]+\eta \left[n\right],$$
where X[n] is the value of *X* in the nth slot, $a_{i}$ and $b_{i}$ are the parameters of the regression, which quantify the relative importance of the past values, *k* is the order of the model, and ϵ[n] and η[n] are the errors of $H_{0}$ and $H_{1}$, respectively. If the knowledge of the past of *Y* cannot contribute additional information about $x_{n}$, $H_{0}$ will have less error variance ϵ[n] than $H_{1}$.

We denote the squared-sum residuals of $H_{0}$ as $RSS_{0}=\sum_{i=1}^{T}{\left(\varepsilon \left[i\right]\right)}^{2}$, and similarly, the squared-sum residuals of $H_{1}$ as $RSS_{1}=\sum_{i=1}^{T}{\left(\eta \left[i\right]\right)}^{2}$, where T=L−k and *L* is the number of data points. Therefore, the F-statistic is given by (Equation 16),(16)$$G_{Y\to X}=\frac{(RSS_{0}-RSS_{1})/k}{RSS_{1}/\left(T-2k-1\right)}∼F\left(d_{1},d_{2}\right),$$
where $d_{1}=k$ and $d_{2}=T-2k-1$. $H_{0}$ will be rejected if $G_{Y\to X}$ is greater than the critical value of F-distribution. The connectivity of two nodes will be existed if the time series of them display Granger causality. We acquired all the connectivity of WSN after deducing Granger causality among all nodes.

## 3. Experiment and Analysis

The console object specifies the number of communication nodes, randomly generates the positions of communication nodes within a 100 × 100 rectangular area, and stipulates that the distance between two nodes cannot be less than a certain threshold. All communication nodes have the capability to transmit information within the specified range. Additionally, it defines the connectivity, creating channels with ten frequency bands in each one. The console is also responsible for allocating channels between communication nodes and reallocating frequency bands after frequency hopping.

### 3.1. Experimental Setup

Considering the small error δ of dichotomy in a known WSN, the result of fitting the coordinate error with a Gaussian mixture model is shown in Figure 7, where the vertical axis represents the error on the y-axis, and the horizontal axis represents the error on the x-axis. The two Gaussian distributions are denoted by two sets of contour lines.

We deployed the parameter to extract local information with self-adaptive error distribution. The comparison of the local information extraction error obtained through the maximum likelihood function and the local information extraction error obtained through trilateration is shown in Figure 8. The horizontal axis represents the node number. The values of the red bars indicate the distance between the estimated coordinates obtained through trilateration and the true coordinates. The values of the blue bars represent the distance between the estimated coordinates applied maximum likelihood function and the true coordinates. It is evident that the overall blue bars are lower than the red bars, which means that extract PI with the maximum likelihood function significantly reduces the error.

The console generates the coordinates of 90 communication nodes and the connectivity, as shown in Figure 9a.The visualization of the topology matrix is shown in Figure 9b, whose transverse and longitudinal axis represent the node numbers and a white block is displayed when there is a connectivity between two nodes and a black block is displayed when there is no connectivity.

The console generates channels with each channel having 10 frequency bands, and each frequency band corresponds to a different Signal-to-Noise Ratio (SNR). The corresponding relationships are shown in Table 1. The console also allocates channels and frequency bands to each pair of communication nodes as shown in Table 2.

EVE draws interfering circles around the transmitting and receiving nodes. After interfering all communication nodes, the identified topology matrix visualization is shown in Figure 9c.

**Figure 7 sensors-25-01347-f007:**
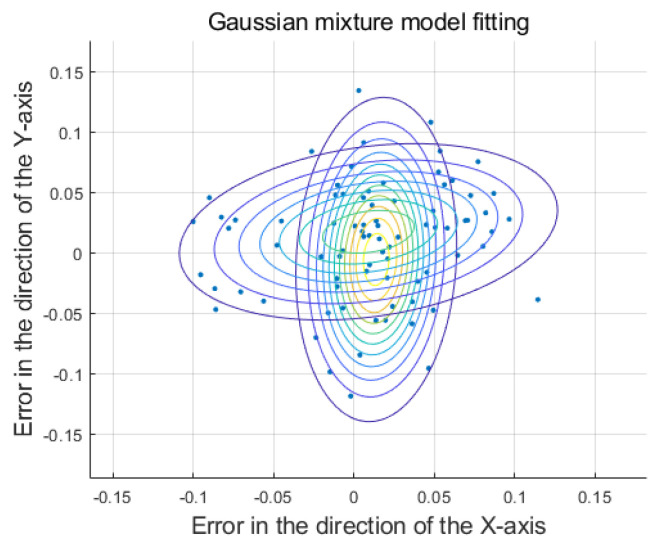
Gaussian mixture model fitting error.

**Figure 8 sensors-25-01347-f008:**
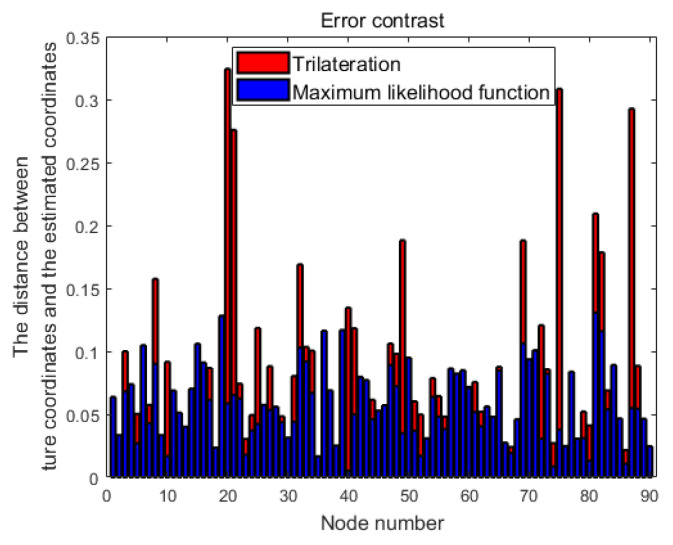
Comparison of PI errors.

In order to reduce the error, we extract PI with the maximum likelihood function. Deploying the extra effective information, we can identify the more precise topology, as shown in Figure 9d.

As time goes on, the topology of WSN changes, whose matrix visualization is shown in Figure 9e. We utilized the BIC model [31] to determine the maximum lag order of the time series and deployed the Granger causality test to infer whether there is a connectivity between two nodes. The identified topology matrix is shown in Figure 9f.

**Figure 9 sensors-25-01347-f009:**
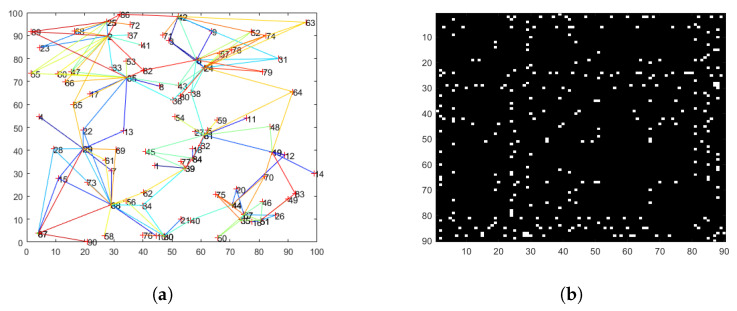
Topology of WSN. (**a**) Network topology diagram. (**b**) Initial topology matrix. (**c**) Topology matrix without modifying. (**d**) Topology matrix after modifying. (**e**) Dynamic topology matrix. (**f**) Dynamic topology deduced by passive interception.

### 3.2. Comparison Result

Based on similar framework AINL also proposed interference strategy. However, the author’s simulation experiments only considered the low noise of interference strategy, and the number of simulations was relatively small. When reproducing the experiment, we increased the noise and continuously changed the positions of the communication nodes to obtain more convincing simulation results.

The number of communication nodes can be regarded as an important indicator of wireless sensor networks scale. We compared the accuracy rates of AIPI, AINL, and passive interception for network topology identification under different network scales and physical layer communication modes. We changed the network scale and the communication mode of the physical layer with Time Division Multiple Access (TDMA), Frequency Division Multiple Access (FDMA), Orthogonal Frequency Division Multiplexing (OFDM), and Code Division Multiple Access (CDMA). The accuracy rates of the identified topology are shown in Figure 10a, Figure 10b, Figure 10c, and Figure 10d, respectively. These four simulation diagrams are obtained by taking the average five repetitions, in order to avoid contingency. It can be seen from Figure 10 and Table 3 that when facing different network scales and physical layer mode, the accuracy rate of our method for network topology identification is above 99%. It has an obvious advantage over the AINL method especially in small-scale networks. The WSN network is sparse, with most nodes not being connected. As the number of nodes increases, the number of non-connected relationships far exceeds the connected ones. AINL method has low positioning accuracy, making it difficult to distinguish between different nodes, which leads to a significant drop in accuracy when dealing with small-scale networks. The passive interception method is generally inferior to the other two methods. This is because passive interception depends on the accuracy of packet decoding. When the channel conditions are poor, the source or type of data packets cannot be accurately decoded, resulting in a low accuracy rate of topology identification. The impact of noise on the four communication modes is similar. Taking OFDM as an example, we define the network size as 40 and conduct topology identification simulations under different SNR levels. As shown in Figure 11, with the increasing of SNR, the accuracy of passive interception, including the Granger causality test and Hawkess process, shows an upward trend. AINL shows an steady performance in different SNR, while AIPI shows an better performance. We considered using four different methods to perform topology identification on the same network five times. As shown in Figure 12, with the increase of the network scale, the power consumption of four methods show the same trend. Given the premise that topology inference requires successfully deciphering non-cooperative network packets, passive interception only requires signal sensing to infer the topology, resulting in the lowest energy consumption. AINL, on the other hand, conducts active interception in every topology identification process, leading to the highest energy consumption. In contrast, AIPI performs active interception only during the initial identification process and utilizes the acquired information for passive interception in subsequent topology identification processes, resulting in moderate energy consumption.

## 4. Conclusions

Even if the network nodes are in a disconnected state, these four methods can still perform real-time identification of the network topology, and AIPI has a higher accuracy rate. The simulation experiments indicate that AIPI combine active and passive interception can inference topology of WSN with high accuracy rates. Active interception is suitable for any protocol but consumes large amounts of energy, and passive interception consumes less energy but is only suitable for the specified protocol. AIPI balances consumption and applicability properly.

The system generates communication nodes within a certain rectangular area, but when the number of communication nodes is excessive, the system’s local information extraction error is relatively large. However, establishing a mixed Gaussian model requires a larger dataset to avoid randomness. When the number of nodes is small, the Gaussian mixture model cannot fit the errors well, which also means that the correction of errors does not achieve a satisfactory effect, leaving considerable room for improvement in error correction.

## Figures and Tables

**Figure 1 sensors-25-01347-f001:**
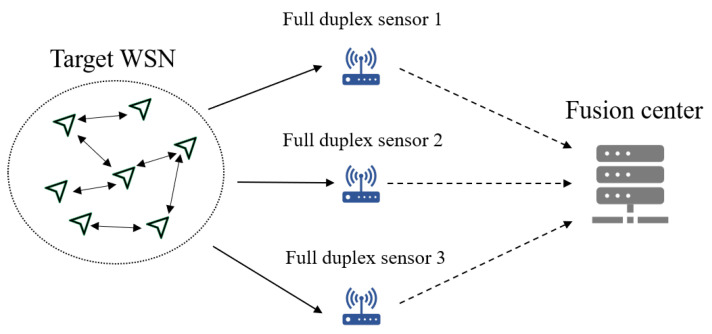
The system model of topology sensing.

**Figure 2 sensors-25-01347-f002:**
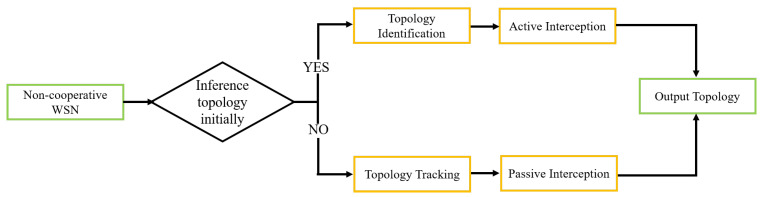
Overall framework of AIPI.

**Figure 3 sensors-25-01347-f003:**
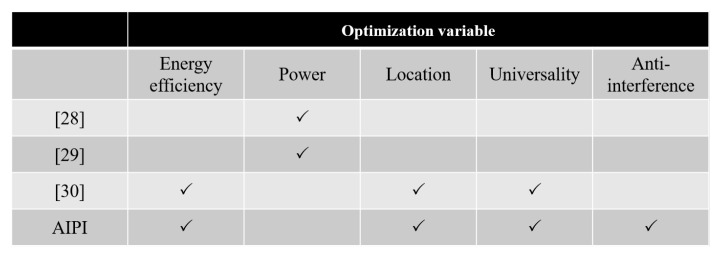
The comparison of AIPI with existing works on topology inference.

**Figure 4 sensors-25-01347-f004:**
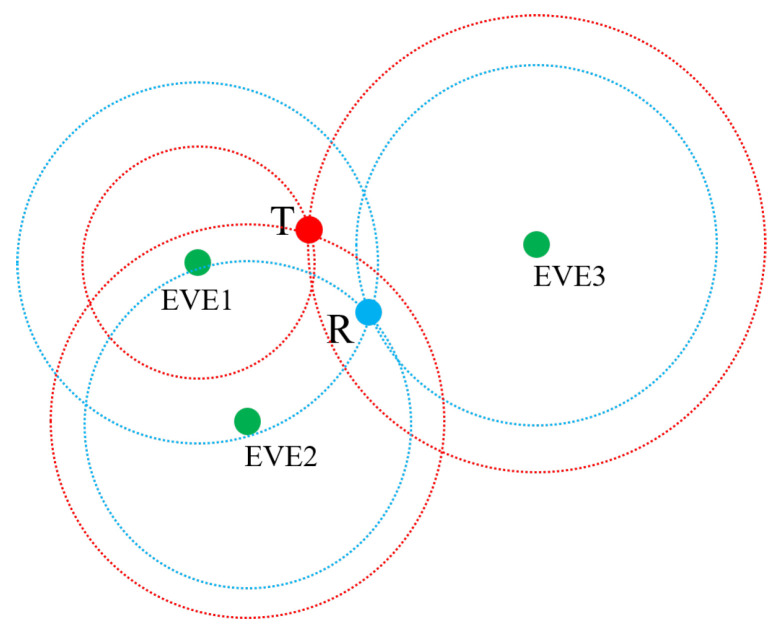
Full duplex sensors extract transmitter and receiver position information.

**Figure 5 sensors-25-01347-f005:**
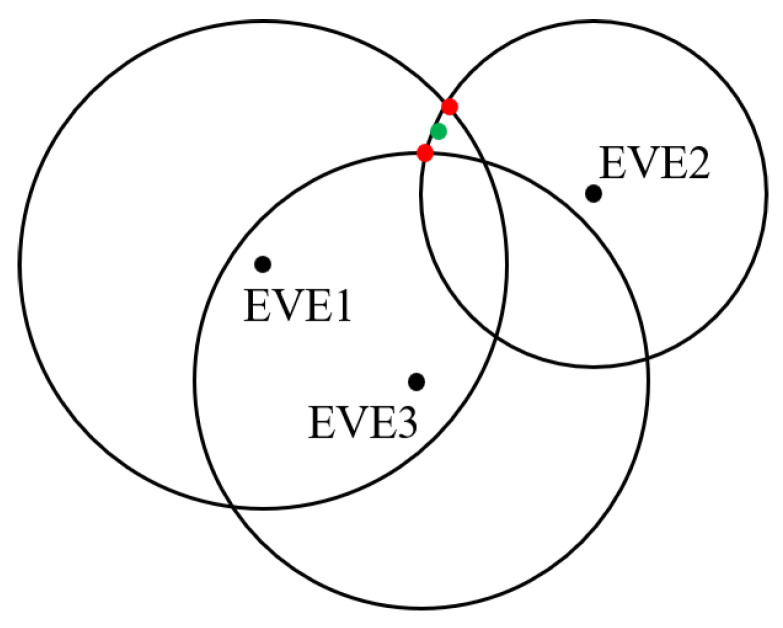
The error from trilateration.

**Figure 6 sensors-25-01347-f006:**
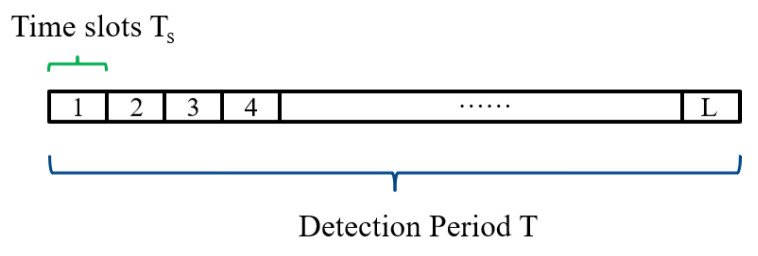
Schematic diagram of the time series data.

**Figure 10 sensors-25-01347-f010:**
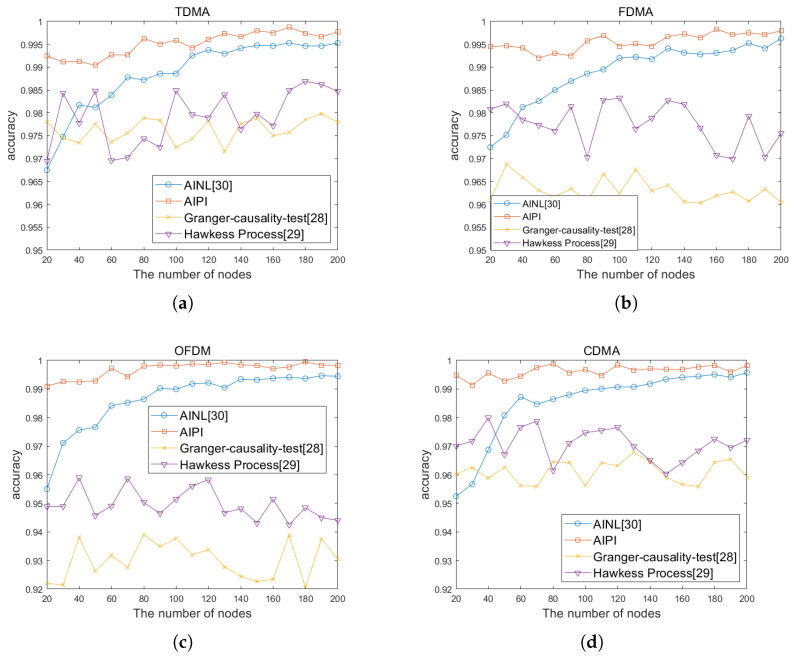
The accuracy rates of the identified topology with different communication modes: (**a**) TDMA, (**b**) FDMA, (**c**) OFDM, and (**d**) CDMA.

**Figure 11 sensors-25-01347-f011:**
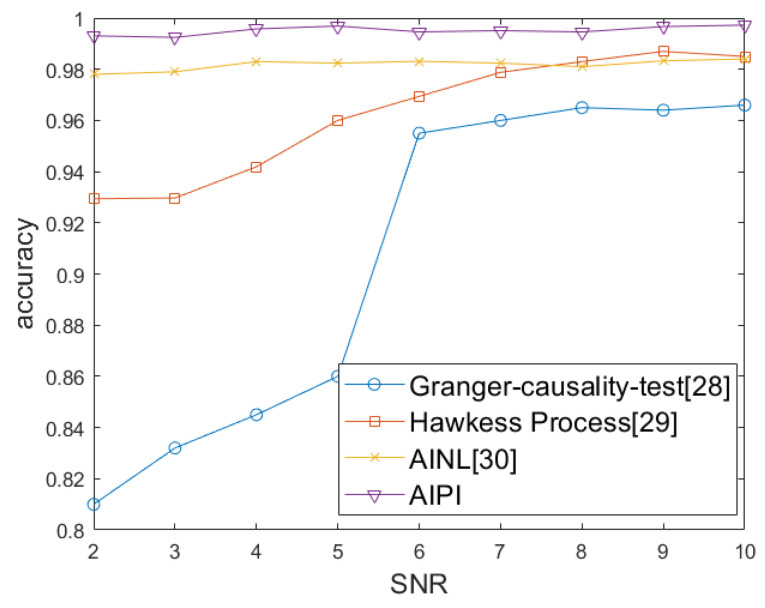
The accuracy rates of the identified topology with FDMA based on different SNRs.

**Figure 12 sensors-25-01347-f012:**
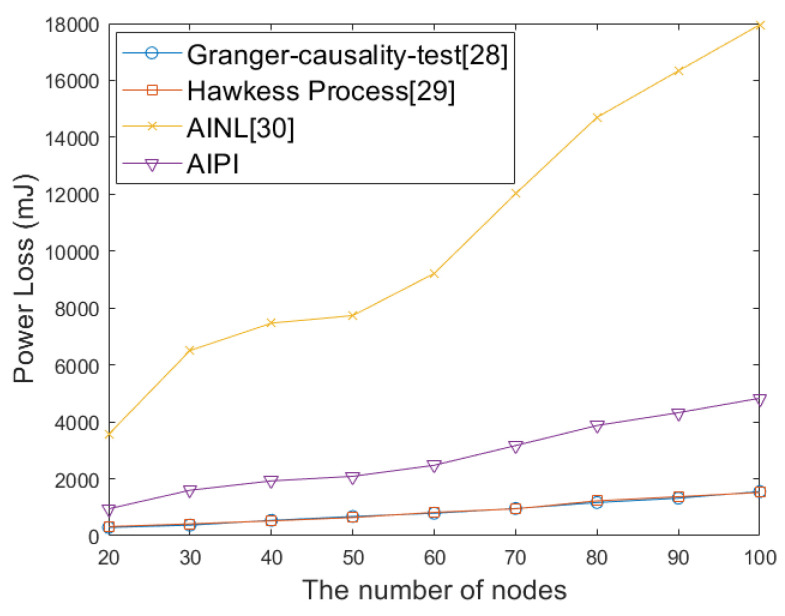
Power loss of four methods with different network scales for five-time identification.

**Table 1 sensors-25-01347-t001:** Channel-band-SNR allocation chart.

channel	1	1	1	1	1	1	1	1	1	1
band	1	2	3	4	5	6	7	8	9	10
SNR	114	99	90	79	123	129	178	30	172	10

**Table 2 sensors-25-01347-t002:** Channel and band of the communication node.

Transmitting Node	1	2	3	3	3	4
Receiving Node	39	25	6	24	42	29
Channel Number	7279	5483	1456	3665	3027	7904
Band Number	5	5	7	6	3	3

**Table 3 sensors-25-01347-t003:** Comparison of the average accuracy of different methods with various network scales.

Methods	Accuracy				
	15 Nodes	50 Nodes	100 Nodes	150 Nodes	200 Nodes
AIPI	99.32%	99.20%	99.63%	99.73%	99.80%
AINL	96.18%	98.03%	99.00%	99.35%	99.54%
Hawkess Process	96.45%	97.06%	97.40%	96.57%	97.13%
Granger causality test	95.53%	95.73%	95.72%	95.51%	95.69%

## Data Availability

The data presented in this study are available on request from the corresponding author (due to privacy).

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
