# Peer review of "AIPI: Network Status Identification on Multi-Protocol Wireless Sensor Networks"

_sensors, 2025, doi:10.3390/s25051347_

Round 1
Reviewer 1 Report
Comments and Suggestions for Authors
1. It is not clear how the reported approach compares against similar existing solutions.
2. Therefore, it is recommended that a separate sub-section is added, which compares the presented model against 3-4 of the most relevant similar approaches. This analysis should highlight the advantages, drawbacks, while the assertions should be sustained with experimental data and/or consistent conceptual remarks.
3. The English language should be improved through, at least, one round of proofreading.
4. The mathematical apparatus should be explained in a more detailed manner, and properly related to its algorithmic and real-world programmatic relevance.
Author Response
Thank you for your comments, please see the attachment.

Reviewer 2 Report
Comments and Suggestions for Authors
please see the attached pdf file

Author Response

(The authors gave the same response as above.)

Reviewer 3 Report
Comments and Suggestions for Authors
1. In the abstract, "life time" ==> lifetime?
2. ", reducing ..." ==> AND reducing interference
3. In line 4, capitalize the meaning of your acronym.
4. line 6: interception ==> interceptions.
5. line 7: ==> , thus, infer their ...
6. I have already seen so many grammatical errors until line 7. I will stop pointing them out from here on and I encourage the authors to proofread their work diligently. There are too many errors after line 7.
7. Before line 57, I did not see what passive interception does. Active interception was discussed in a more detailed manner. One comment here is that the interceptions were not discussed towards building up your problem solution to your proposed way. Also, It would be better to provide a figure showing your WSNs with passive and active interceptions.
8. I did not understand before line 57 how your solution was different from active and passive. Is this a mere combination of two methods?
9. Your proposed AIPI method was not discussed in the introduction. Provide some discussions on how this is different before you proceed with your contributions.
10. What does Fig. 1 discuss? What do you mean by the decision block there?
11. In Fig. 2, what or where is PI? What do the red and blue circles signify? Provide a better caption.
12. In Fig. 3, what are the colored dots there?
13. Why equate equation 10 to zero? to maximize?
14. In section 2.2, you mentioned that after active interception, you apply passive. This is not the one shown in Fig. 1, right?
15. Section 2 is not discussed comprehensively. I do not see any novelty in the work as they claim. They are just trying to combine to methods. Re-write or explain your work more clearly.
16. Why does Fig 7 come first in the discussion before Figs 5 and 6? Do referencing chronologically.
17. Figure 7 does not provide a clear explanation of their results. What do the readers need to know here? Explicitly state your findings.
18. How does Figure 7f show the topology? Shouldn't it be presented like Figure 7a showing connections among nodes?
19. Figure 8 is deceiving. Your y-axis values are close. So is a 0.03 accuracy difference very big for the novelty of your work? If implemented in real-life, will the cost of introducing your framework even worthy of this small difference? Please emphasize.
20. Are there other performance metrics that the authors can use to present their work in a better way? Please provide more tests and comparisons.
Comments on the Quality of English LanguageThe paper needs an extensive proofreading. I stopped placing comments about their style in line 7.
Author Response

(The authors gave the same response as above.)

Round 2
Reviewer 2 Report
Comments and Suggestions for Authors
I am satisfied with how the authors addressed my recommendations. I recommend that the paper be accepted in its current form.
Reviewer 3 Report
Comments and Suggestions for Authors
Thank you for addressing my comments/suggestions.
Comments on the Quality of English LanguageI hope you did check the grammar and construction.